# Mechanistic Insights and Therapeutic Delivery through Micro/Nanobubble-Assisted Ultrasound

**DOI:** 10.3390/pharmaceutics14030480

**Published:** 2022-02-22

**Authors:** Shirui Lu, Pengxuan Zhao, Youbin Deng, Yani Liu

**Affiliations:** Department of Medical Ultrasound, Tongji Hospital, Tongji Medical College, Huazhong University of Science and Technology, Wuhan 430030, China; d202182134@hust.edu.cn (S.L.); zhaopengxuan@tjh.tjmu.edu.cn (P.Z.)

**Keywords:** ultrasound, micro/nanobubbles, drug delivery, membrane permeability

## Abstract

Ultrasound with low frequency (20–100 kHz) assisted drug delivery has been widely investigated as a non-invasive method to enhance the permeability and retention effect of drugs. The functional micro/nanobubble loaded with drugs could provide an unprecedented opportunity for targeted delivery. Then, ultrasound with higher intensity would locally burst bubbles and release agents, thus avoiding side effects associated with systemic administration. Furthermore, ultrasound-mediated destruction of micro/nanobubbles can effectively increase the permeability of vascular membranes and cell membranes, thereby not only increasing the distribution concentration of drugs in the interstitial space of target tissues but also promoting the penetration of drugs through cell membranes into the cytoplasm. These advancements have transformed ultrasound from a purely diagnostic utility into a promising theragnostic tool. In this review, we first discuss the structure and generation of micro/nanobubbles. Second, ultrasound parameters and mechanisms of therapeutic delivery are discussed. Third, potential biomedical applications of micro/nanobubble-assisted ultrasound are summarized. Finally, we discuss the challenges and future directions of ultrasound combined with micro/nanobubbles.

## 1. Introduction

It is well known that the oral administration and intravenous injection of most drugs can produce significant systemic side effects. Therefore, it is necessary to control the release of drugs at targeted sites in a spatially and temporally controlled manner to reduce the adverse systemic toxicity [1]. In recent years, a large number of studies have been published on the development of stimulus-responsive drug delivery systems which can precisely control drug release in response to endogenous or exogenous stimuli [2,3]. However, for endogenous stimuli such as temperature and pH, it can be hard to precisely control release sites and rates due to the heterogeneous disease environment. Exogenous stimuli such as heat and light can manually control activation properties but may cause tissue damage, and the depth of penetration may not be enough to trigger the release of drugs into deep tissue [4].

Ultrasound (US), as a most significant exogenous stimuli [5], has many advantages including noninvasive, real-time imaging and being precisely focused and controlled [6,7]. In addition, ultrasonic waves have a unique physicochemical property to increase the permeability of drugs and release drugs through the biological barrier [8,9]. Therefore, micro/nanobubble-assisted US has been developed not only for imaging but also for therapeutic delivery [10].

Microbubbles (MBs) are spheres with diameters of 1–10 μm, composed of a core filled with air or other gases, whereas the outer shell is composed of lipids, polymers or proteins [11]. The unique advantage of MBs is that they can not only serve as contrast agents for imaging but also as suitable vehicles for the loading of therapeutic materials and precise delivery at targeted sites [12]. However, MBs only remain in the circulation until dissolving due to the micron-scaled sizes, so the MBs cannot passively extravasate to deep tissues [13]. To overcome this limitation, nanobubbles (NBs) have been investigated for ultrasound-triggered drug release outside the bloodstream, which have been known to exist in commercially available MB formulations such as Definity^®^ (phospholipid-coated perfluoropropane-filled bubbles, mean size 1.1–3.3 µm, each mL contains a maximum of 1.2 × 10^10^ perflutren lipid microspheres) [14]. The other currently marketed product is Optison^®^, microbubbles stabilized by human serum albumin with perflutren (mean size 3.0–4.5 µm, each mL contains 5.0–8.0 × 10^8^ protein-type A microspheres). However, the disadvantage of NBs is that high-quality ultrasound imaging cannot be obtained because small bubbles will reduce the acoustic response. One of the strategies for manufacturing NB imaging is to adjust and modify the shell composition of NBs to increase their echogenicity [15,16,17,18]. Although NBs provide US contrast enhancement at frequencies below their resonance and hence provide promise for diagnostic use, they are still in the preclinical stage [19]. Recently, preclinical investigations of the combined therapeutic use of ultrasound and micro/nanobubbles have been widely conducted in the field of cardiovascular diseases [20], solid cancers [21,22] and so on.

In this review, we classified and discussed the structure and generation of micro/nanobubbles, in addition to summarizing the US parameters and the mechanisms of therapeutic delivery. Furthermore, we highlighted the potential biomedical applications of micro/nanobubble-assisted ultrasound. Finally, we discussed the challenges and future directions of micro/nanobubble-assisted ultrasound (Figure 1).

## 2. Micro/Nanobubbles Structure

Micro/nanobubbles consist of two main components with different physicochemical characteristics: inner core and outer shell (Figure 2a). Considering the essential requirements of safety, biocompatibility, biodegradability and regulatory admit, the shell affects the mechanical elasticity and mainly consists of lipids, surfactants, polymers, proteins or polyelectrolyte multilayer [23,24]. Inner core determines the acoustic response and contains air, oxygen, sulfur hexafluoride or perfluorocarbons [25,26]. In order to enhance the delivery efficiency and decrease non-specific toxicity, bubbles are widely modified with special targeting ligands on the shell surface [14].

### 2.1. Outer Shell and Inner Core

Lipids and polymers are the most commonly used shell components to improve stability and prevent gas dissolution and bubble coalescence. The shell composition can affect inner gas exchange from the core to the external medium, further influencing the bubbles’ half-life. Moreover, shell thickness and elasticity can significantly influence the bubbles’ stability, and the viscoelastic properties of the shell can influence the bubbles’ response in an acoustic field [27,28]. Bubbles exposed to US lead to rapid contraction and expansion, causing the shell to oscillate or even rupture. Therefore, when the pressure change is small, the soft shell will break, but the hard shell will not. Shells that are too rigid will reduce the echo effect, while a thin shell will promote gas diffusion [24].

Since the encapsulated core gas has a significant effect on the stability of bubbles, one way to increase stability is to use insoluble gases. However, original used air is limited by low resistance to water permeation. The novel developed core composition contains a high molecular weight, low solubility gas such as perfluoropropane (C3F8) and sulfur hexafluoride (SF6) [29].

### 2.2. Therapeutic Cargo Loading

Micro/nanobubbles commonly use three possible methods to load therapeutic cargo [30]. First, micro/nanobubbles encapsulate cargo within the inner core. Lipophilic drugs can be directly dissolved in perfluorocarbons. Studies have shown that combining with a cosolvent or adding oil to the core can promote lipophilic drugs dissolution [31]. Besides, micro/nanobubbles can carry special gases such as hydrogen (H_2_), nitrogen (N_2_), oxygen (O_2_) or nitric oxide (NO) in their core, which can be useful for biomedical applications by influencing physiological and pathophysiological processes [32,33]. Second, drugs can be loaded inside or under the shell through an electrostatic connection [34]. For example, hydrophobic drugs can be directly loaded into the lipid shell. However, due to the relatively thin mono-layer shell, the drug loading efficiency is not high enough. Although charged materials such as nucleic acids or doxorubicin can be easily electrostatically coupled with anionic/cationic shell, it is limited by the uncontrol release of drugs [35]. To overcome this limitation, multilayer systems through chemical binding to the bubbles’ shell surface have been explored. For example, a multilayer system coating bubbles with DNA and polymers was used to continually bind oppositely charged polyelectrolytes onto the surface of shells [36]. The polymer layer was adopted to prevent DNA from enzymatic dissolving. The special multilayer technique showed a significant increase in the amount of drug loading. The third approach is encapsulating drugs into a nanometer material which is then linked to the bubbles’ shell. Using ligand−receptor interactions can also achieve drug loading on the shell.

The versatile structure of micro/nanobubbles has also been explored for co-delivery of two or more disparate drugs [37]. For example, Cavalli et al. designed a kind of dextran NB in which positively charged cisplatin was loaded within the bubble shell, while dissoluble doxorubicin is reserved in the gas core [38]. When irradiated by US, the bubbles carried with two different drugs showed a significantly improved therapeutic efficacy against tumor cells.

### 2.3. Surface Modification

Targeted bubbles have gained considerable attention because of their advantage in achieving a high degree of accumulation of therapeutic drugs in targeted tissues by adjusting drug pharmacokinetics and biodistribution. Depending on whether the surface ligands bind, targeted delivery can be classified into a passive or active target [39]. The passive target is characterized by passive orientation toward the target site without the ligand used, while the active target is achieved with the conjugation to specific ligands. The ligands enable recognizing specific receptors in the target site, thus enhancing tissue selectivity and reducing nonspecific toxicity through surface modifications [40]. For instance, Jiang et al. used Herceptin-targeted NBs and found that these NBs could efficiently penetrate into Her-2-expressing tumors with less toxicity [41]. Another typical example is that glycosylation has been applied for both macromolecular and liposomal carriers for cell-selective drug targeting, the receptor-mediated targeting by glycosylation has been used for therapy with nucleic acids [42].

## 3. Micro/Nanobubbles Generation

Micro/nanobubbles are either spontaneously formed by US (endogenous) or externally generated (exogenous). The exogenous pathway includes directly generating bubbles and suppling as intermediate forms such as emulsions which are transformed into bubbles upon US triggering. Techniques that can be used to produce exogenous micro/nanobubbles include sonication, emulsion, mechanical agitation, laser ablation and so on [43]. The common procedure is to use high-intensity ultrasound to disperse liquid or gas into a suspension of coating ingredients, thus emulsifying the liquid or gas to form bubble/droplet suspension, which can automatically adsorb onto coating material such as proteins or surfactants. In addition to bubbles produced directly, such as MBs, NBs can also be generated by filtration, floatation, centrifugation and condensation of MBs, which decrease in size due to the dissolution of interior gases by the surrounding liquid [44].

### 3.1. Endogenous Generation

Because of the presence of gaseous pockets in normal tissues, bubbles can be generated endogenously by activating physiological gaseous pockets using ultrasound at an intensity greater than the cavitation threshold [45]. A study showed that endogenous MBs generated by low-frequency ultrasound irradiation can help drug accumulation in the gastrointestinal tract, making it suitable for treating gastrointestinal diseases [46].

### 3.2. Exogenous Generation

Compared to endogenous bubbles, exogenous bubbles have a lower cavitation threshold. In general, there are three ways of generating bubbles. The most common class uses the compression of the air stream to dissolve air into liquid, which is subsequently released through a specially designed nozzle system, to nucleate small bubbles as potentially nanobubbles, based on the cavitation principle. The second class uses power ultrasound to induce cavitation locally at points of extreme rarefaction in the standing ultrasonic waves. The third class uses an air stream delivered under low offset pressure [47]. The most common disadvantage of directly generating bubbles is the low efficacy of tissue targeting due to the short circulation time. To overcome this limitation, bubbles can be designed to stabilize in the intermediate form of emulsions. The droplet emulsions can undergo acoustic droplet vaporization (ADV) process to change into gas bubbles approximately five to six times larger in diameter after being irradiated by US [48]. The pressure required to convert droplets into gas bubbles depends on the characteristics of droplets such as shape and size [49]. For example, emulsions made of perfluorocarbon (PFC) express ideal inertness and biocompatibility with increased circulation time and larger loading capability compared to PFC MBs [50]. Studies have shown that smaller droplets are more stable and require higher negative pressure amplitudes [51,52]. Besides, emulsions are smaller than MBs and can potentially extravasate out of blood vessels. When functionalized with ligands, emulsions can increase targeting specificity [53]. In addition, the alliance of nanoemulsions with nanoparticles can significantly improve therapeutic efficacy. For example, PFC nanoemulsions associated with silica nanoparticles show effective strategies for improving cancer treatment [54].

## 4. US Parameters

Therapeutic efficacy can be significantly promoted by the optimizing of US parameters. Therefore, standardization of US parameters is significant to increase efficiency and minimize unwanted damages [55]. The frequency, intensity and mechanical index of irradiated US have been proved to be the main influence parameters [56,57]. Any one of these ultrasound parameters will largely affect drug delivery processes by influencing bubble−cell interactions. It is valuable to modulate responses by changing ultrasound conditions through estimating the impact of ultrasound settings on the physiologic process involved in ultrasound-induced drug delivery [58].

Upon applying US, micro/nanobubbles grow under negative pressure, whereas they contract under the positive pressure of ultrasound waves. Therefore, US frequency has an important influence on therapeutic efficiency. Therapeutic US typically requires lower frequencies to penetrate deeply into tissue and induce cavitation compared to diagnostic US [59]. The cavitation behavior of bubbles at a certain frequency will mainly depend on their size, as bubble response will be higher around their resonant radius [60]. Most studies report an ultrasound center frequency of around 1 MHz because it roughly matches the resonance frequency of the majority of bubbles (volume-weighted) in standard microbubble formulations (±3 μm in size) [61].

US intensity is a measurement evaluating heat effect. It must be limited within a safety zone to avoid irreversible thermal damage to cells [62]. The therapeutic US typically applies between 0.3 to 3 W/cm^2^ to maximize drug delivery efficiency and minimize damages to normal tissues [63]. Different from high-intensity focused ultrasound (HIFU) which can thermally ablate tissues via hyperthermia in different carcinomas at 1000 W/cm^2^ [64], low-intensity ultrasound (US), defined as therapeutic US with a relatively lower intensity than HIFU, has a great potential in apoptosis therapy for cancer and can be relatively easily applied [65].

The mechanical index is calculated as the ratio of peak negative pressure to the square root of acoustic center frequency. It mainly reflects the mechanical effects [33]. For example, while bubbles play a significant role in the process of US-induced blood–brain barrier (BBB) opening, the level of cavitation involved in the bubble−ultrasound interaction must be evaluated to guarantee the BBB-opening level and quality. With the mechanical index as the major indicator in gauging inertial cavitation activity, it would be valuable to understand the roles of cavitation sources on ultrasound-induced BBB-opening [66]. The effective methods of increasing the mechanical index and promoting cavitation serve to increase pressure amplitude or decrease frequency [67]. For activation of micro/nanobubbles, the mechanical index used is typically in the range of 0.2~1.9 [63].

Moreover, it is reported that the ultrasound pulse length can have a major impact as well. Compared with long ultrasound pulses (ms to s), it seems that very short pulses (few µs) might be more efficient in combination with high acoustic pressures [68]. Karshafian et al. reported that cell permeability increased and viability decreased with increasing peak negative pressure, pulse repetition frequency, pulse duration and insonation time and with decreasing pulse center frequency. However, cell permeability and viability did not correlate with bubble disruption. The results indicated that ultrasound exposure parameters can be optimized for therapeutic sonoporation and that bubble disruption is a necessary but insufficient indicator of ultrasound-induced permeabilization [69].

## 5. Mechanisms of Micro/Nanobubble-Assisted Drug Delivery

The main potential mechanisms of micro/nanobubble-assisted drug delivery include a cavitation effect, sonoporation effect, acoustic radiation force, acoustic streaming and thermal effect. These mechanisms interact to promote drug penetration through the vascular wall or cell membrane, producing a series of physical and chemical effects (Figure 2b,c).

### 5.1. Cavitation Effect

The major driving force for US-guided bubble bursting was ultrasonic cavitation, which can increase the bubbles’ permeability into tissues and cell membranes [70]. Defined as the formation, pulsation and collapse of bubbles, cavitation is caused by the interaction of US energy and bubbles. It promotes the expansion or contraction of micro/nanobubbles [60]. Different acoustic parameters lead to bubbles generating different oscillation outcomes. At low ultrasound intensities, bubbles oscillate as stable cavitation (SC) in a stable movement. In the expansion stage of the bubble core, there is a net inflow of gas. The bubble expands until it reaches the resonant size and oscillates with low amplitude in the linear direction. The regular oscillation in SC can echo the ultrasonic wave utilized by ultrasound imaging. In contrast, when irradiated by ultrasound at high intensities, bubbles oscillate through the inertial cavitation (IC) that accompanies explosive growth, collapsing and oscillating in an asymmetric non-linear manner [71]. The differences between SC and IC are the degree of bubble deformation and physical effects. SC tends to produce micro-streaming, but IC tends to produce fluid jetting, shock waves or free radicals. In addition to changing the permeability of the blood-vessel wall, the cavitation effect can also enhance the permeability of the cell membrane and promote locally releasing therapeutic cargo [72]. When oscillating bubbles approach the cell surface, they exert pressure and shear forces on the cell membrane, promoting cell membrane fragmentation and thus increasing intracellular drug absorption [73].

### 5.2. Sonoporation Effect

Cavitation vibration can cause a mechanical effect, which in turn induces the opening of the connections between endothelial cells, resulting in the formation of instantaneous micropores in the cell membrane with a diameter of several to 150 nm. This process is called the sonoporation effect [74]. This process leads to the diffusion of surrounding molecules into the cytosol. The mechanisms contributing to sonoporation are categorized according to three ultrasound settings: (i) low-intensity ultrasound leading to stable cavitation of bubbles, (ii) high-intensity ultrasound leading to inertial cavitation with bubble collapse and (iii) ultrasound application in the absence of bubbles [74]. Using low-intensity ultrasound, the endocytotic uptake of several drugs could be stimulated, while short but intense ultrasound pulses can be applied to induce pore formation and the direct cytoplasmic uptake of drugs. The sonoporation effect has been extensively utilized to facilitate targeted delivery [15]. For example, Huebsch et al. found that the sonoporation effect occurs under ultrasound irradiation through the hydrogel system, which leads to temporary and large-dose drug release and increases the toxic effect of chemotherapy drugs [75].

### 5.3. Acoustic Radiation Force

When acoustic waves propagate to the medium, mechanical force can be generated due to momentum transfer. This mechanical force is called acoustic radiation force (ARF), which includes primary forces and secondary forces [76]. The primary forces act on single particles [77], but secondary forces are the forces that occur between particles [78]. Besides, the primary forces mainly push the particles to migrate in the sound field, while secondary forces mainly cause the aggregation and dispersion of particles [79,80]. In particular, the ARFs that act on the bubbles are also named Bjerknes forces [80]. The consequence of Bjerknes forces is that for small bubbles, they will be collected at pressure maxima and become active there, while large ones will go to pressure minima and become inactive [60]. Bjerknes force has been used to ultrasonically concentrate erythrocytes, DNA and hybridoma cells [60]. Each interaction within bubbles can be explained by different equations. The parameters in the equations mainly include the physical and chemical characteristics of the particles [80].

### 5.4. Acoustic Streaming

Ultrasound produces reflection and scattering in the propagation process, and the reflected or scattered particles produce a force on the surrounding medium. This force is called acoustic streaming, which is divided into bulk streaming and microstreaming. Bulk streaming is the acoustic streaming that leads to moving along the ultrasonic propagation direction, while microstreaming is the local force around the bubble with an unfixed direction. This acoustic streaming force can not only promote the migration of bubbles [81], especially in blood vessels [82], but also the release of drugs from the carrier into the target tissue [83]. The fluid microjet can enhance mechanical stress on the cell membrane and lead to transient disruption to promote drugs delivering into the cell cytosol [74]. Acoustic streaming plays an important role especially in low-frequency sonophoresis [84].

### 5.5. Thermal Effect

The interaction of ultrasound and micro/nanobubbles can produce mechanical and thermal effects [85]. As the sound wave propagates through the medium, the interaction force between the sound wave and the medium gradually transforms the sound energy into heat energy [10]. In micro/nanobubble-assisted ultrasound strategy, localized heating of the targeted tissue can improve the therapeutic efficacy and prevent additional high-temperature damage to normal surrounding cells [9]. A previous study showed that the thermal effect can effectively stimulate drug release when the temperature rises above normal physiological temperature [86], but ultrasound with too-high energy may cause ablation damage to normal tissue [87].

## 6. Biomedical Applications

The special advantage of micro/nanobubbles combined with US makes them conducive in various therapeutic applications such as in the treatment of tumors, diabetes mellitus, atherosclerosis, myocardial infarction, neurodegenerative disease and other advanced disease areas.

### 6.1. Tumor Therapy

Due to the high morbidity and mortality of cancer, it is very important to make efforts to detect effective treatment strategies [88]. At present, the preferred clinical treatments for cancer are surgery, chemotherapy and radiotherapy [89]. Although these have a certain therapeutic effect, they cannot target the tumor tissue, thus causing toxic and side effects on the normal tissue. Due to the low local drug concentration in the tumor, the risk of long-term metastasis is still high [90]. Instead, micro/nanobubbles enable antineoplastic agents releasing at local tumor sites, overcoming the shortages of traditional treatment strategies [91]. For example, using novel bifunctional nanodroplets as smart carriers, Gao et al. designed a new strategy for anticancer drug delivery in vivo (Figure 3A) [92]. Nanodroplets using perfluorohexane (PFH) to fill bubble cores with chitosan/alginate complexes as encapsulation shells were not only affected by endogenous local tumor microenvironment pH, but also stimulated by exogenous ultrasound irradiation, and the two synergistically achieved local drug release. The study showed this novel droplet had an excellent antitumor therapeutic effect, indicating new progress in ultrasound-assisted drug delivery in tumor therapy. In addition, during the phase change of the nanodroplets into bubbles, the difference in acoustic impedance between the bubbles and their surrounding medium was increased, which provides the basis for imaging monitoring for the evaluation of the therapeutic effect. In another study, bubbles were designed with a poly (amino acid) shell encapsulating perfluoropentane (PFP)-pentafluorobutane (PFB) and doxorubicin (DOX) (Figure 3B) [93]. After reaching the tumor site, stimulated by the local acidic microenvironment, it expands and increases in diameter, thereby reducing the cavitation threshold. Combined with the continuous irradiation of low-frequency ultrasound, the air bubbles undergo inertial cavitation, which promotes the release of DOX into the deep tumor site.

Since the tumor microenvironment is often in a hypoxic state, in order to increase the concentration of the drug delivered to the hypoxic site, the use of special cells to load the drug, combined with ultrasound-assisted irradiation, is beneficial to increase the sensitivity of chemotherapeutic drugs to hypoxic lesions. For instance, Huang et al. used monocytes/macrophages to load polymer-shelled bubbles and dox polymer-loaded vesicles, irradiated by ultrasound to control drug release. The results showed the combination of US-assisted bubbles and drug-loaded cells could effectively increase drug delivery to hypoxic tumor sites and improve chemotherapy efficiency [94].

The mechanism of efficient infusion is inseparable from the unique vascular microenvironment of the tumor itself, which is characterized by vascular tortuosity, high fluid pressure, growth-induced angiogenesis and solid hypoxia [95]. The main factors that prevent drugs from reaching the tumor site are abnormal tumor blood vessels and high interstitial pressure [96]. Due to the lack of sufficient connective tissue support in the blood vessels at the tumor site, the vascular walls often form voids of different sizes, which are beneficial for drug delivery and increase the enhanced permeability and retention (EPR) effect [97]. Many current delivery therapeutic systems designed based on the EPR effect have shown very high application value [98,99]. The combination of micro/nanobubbles with other therapeutic strategies such as immunotherapy and photothermal therapy is an effective way to increase the EPR effect of drugs. For example, Li et al. designed a combination of ultrasound-assisted bubble and photodynamic therapy in vivo (Figure 4A) [100]. Using the mechanical shear force generated by inertial cavitation of bubbles could efficiently enhance the sensitivity of light stimulation. The dual exogenous stimulation strategy could not only help to overcome the limitation of low permeability, but also significantly increase drug release concentration. 

Furthermore, micro/nanobubble-assisted ultrasound could overcome the blood–brain barrier (BBB) and release drugs at a deep site, which is beneficial for brain tumor therapy. Huang et al. loaded iron oxide and NB in silicon-based nanometers for intracerebral delivery (Figure 4B) [101]. Iron oxide first converged NBs to the target side under the action of an external magnetic field and then used high-frequency US to destroy the BBB, thereby effectively increasing the distribution concentration of the drug in the brain tissue.

### 6.2. Diabetes Mellitus

As a chronic disease with high morbidity, diabetes mellitus (DM) has impacted the lifestyle of billions of people in the last few decades [102]. According to pathogenesis, DM can be divided into type 1 DM (T1DM) and type 2 DM (T2DM). T1DM is caused by absolute deficiency of insulin, while T2DM is due to insulin resistance. The hyperglycemic status in DM can increase the risk of chronic vascular diseases such as nephropathy, stroke and cardiovascular disease [103]. Conventional therapy strategies include oral or intravenous administration agents [104]. To overcome the disadvantages of traditional administration such as low bioavailability and side effects, micro/nanobubble-assisted ultrasound has been widely explored for whole-body glucose homeostasis. For example, the adiponectin gene was delivered to skeletal muscle employing ultrasound-assisted MBs to improve sensitivity to insulin [105]. A glucose tolerance test in mice showed that gene transfer can increase the expression of adiponectin in skeletal muscle, thereby effectively improving glucose tolerance.

Micro/nanobubble-assisted ultrasound strategy has been explored for early intervention with diabetic complications. For example, Zheng et al. found that the decreased myocardial function of diabetic cardiomyopathy was expected to be improved through fibroblast growth factor (FGF1) treatments [106]. The most significant improvement was found in the FGF1-loaded nanoliposomes combined with US-induced MBs destruction. Another common complication of DM is diabetic nephropathy. Although coenzymeQ10 (CoQ10) has potential value in the early treatment of diabetic nephropathy, its water-insolubility and non-specific distribution limit clinical application. Yue et al. adopted CoQ10-loaded liposomes (CoQ10-lip) associated with US-targeted MBs to therapy diabetic nephropathy rats. Results showed a significant improvement in renal hemodynamics. In addition, the 24 h urinary protein and oxidative stress indexes were decreased, indicating a significant recovery of renal function (Figure 5) [107].

### 6.3. Atherosclerosis

Atherosclerosis is one of the most common phenotypes in cardiovascular diseases. It is characterized by the destruction of the intima of the blood vessel wall, causing the aggregation of inflammatory cells and immune cells and the local release of chemical factors, resulting in the deposition of lipids and fibrin in the arterial intima to form plaques [108]. As the fibers and lipids accumulate, blood vessel walls eventually form plaques that narrow the lumen of blood vessels and eventually induce organic diseases [108]. There are different methods to prevent and treat atherosclerosis such as anti-inflammation drugs, cholesterol-lowering drugs and anti-platelet drugs. To enhance the biological effects of these drugs, US-targeted therapies are developing for the prevention of microvascular obstruction [109]. Because of endothelial dysfunction in atherosclerosis, NBs are superior in passive-targeting than MBs through permeability and retention effects [110].

Sonothrombolysis refers to the dissolution of intravascular thrombus through the process of ultrasound-induced cavitation to disrupt the fibrin mesh [111]. Its mechanisms include thrombus fragmentation and augmented penetration of thrombolytic agents [112]. A systematic review included 35 studies and found that the recanalization rates in the group that conducted sonothrombolysis were higher, suggesting that combined microbubbles and ultrasound is a safe and effective strategy for thrombosis treatment [113].

Another clinical target to monitor and treat atherosclerosis is intraplaque neovascularization. For example, Yuan et al. combined US with intercellular adhesion molecule-targeted MBs loaded with Endostar^®^, which is a typical angiogenesis inhibitor acquired from the alteration of endostatin [114]. An obvious plaque decrease was observed compared with the control group, indicating the underlying application value of MB-assisted US in atherosclerosis therapy.

### 6.4. Myocardial Infarction

As a major burden on society, myocardial infarction (MI) is defined as myocardial ischemic necrosis due to coronary artery occlusion and insufficient blood supply. The main cause of MI is the thrombotic occlusion of a coronary vessel [115]. Since MI can induce profound metabolic and ionic perturbations in the affected myocardium, it is important to prevent the rapid depression of systolic function. Despite progress having been achieved in treatment strategies of MI, the dysfunction of myocardial cells and the constitution of fibrous composition still have the potential to develop congestive heart failure. Therefore, increased blood perfusion and repairment of impaired myocardial cells are important for improving prognosis. The treatment principles of micro/nanobubble-assisted ultrasound are: (1) sonothrombolysis; (2) modulating vascular to enhance perfusion; and (3) repair of damaged cardiac tissue [20].

The mechanism of micro/nanobubble-assisted ultrasound to increase perfusion is through vasodilating vessels and reducing vascular wall stiffness by inertial cavitation-triggered release of endothelial generated vasodilators [116]. The increased blood supply generated during cavitation is beneficial to promote the endothelium and erythrocytes releasing massive adenosine triphosphate. Adenosine triphosphate can accelerate endothelial release of prostaglandin and NO. NO and prostaglandins are metabolized to adenosine, which not only relaxes smooth muscle cells but also reduces inflammation and platelet aggregation [117]. For example, Li et al. adopted a canine coronary microthrombi model and found that ultrasound combined with MB delivery of drugs showed a more significant improvement in myocardial perfusion compared with the control group, and the myocardial function was effectively improved [118].

In addition, using stem cells to repair dysfunctional cardiomyocytes has been reported to hold great promise in the treatment of MI. For example, Kokhuis et al. investigated a new strategy by combining stem cells and MBs [119]. They found that the strategy can effectively repair damaged myocardial tissue and increase heart function. To promote the stem cells homing into the infarcted myocardium, MBs with NO loaded into their core have been used in cell transplantation. For instance, Tong et al. found that mesenchymal stem cells migrated more efficiently, and the capillary density in the US + NO MBs group was significantly higher than that in the control group, indicating the cardiac function was markedly improved [120].

### 6.5. Neurodegenerative Disease

Neurodegeneration diseases are complicated, debilitating disorders with a high incidence. It is very emergent to find new effective treatments to prevent the progression of these diseases [121].

As a common progressive neurodegenerative disease, Parkinson’s disease (PD) is characterized by Lewy bodies formatted in alpha-synuclein-dominated neurons, which are caused by the death of dopaminergic neurons in the substantia nigra pars compacta of the midbrain [122]. Although the main treatment method is to supplement the endogenous source of dopamine, it cannot prevent continuous neuronal degeneration and may ultimately lead to recurrence [123]. Since the combination of micro/nanobubbles with transcranial low-intensity focus US can overcome the BBB, a micro/nanobubble-assisted ultrasound strategy is widely developing as a promising strategy to treat PD [124]. For example, Yan et al. explored a new platform for targeted delivery of curcumin into the mouse brain (Figure 6) [125]. They adopted melt-crystallization approaches to enhance curcumin dissolving in water and induce curcumin-carried lipid-poly (lactide-co-glycolide) (PLGA) NBs (Cur-NBs) by the encapsulation of curcumin within the nucleus of lipid-PLGA NBs in a mouse model. Results showed that Cur-NBs combined with low-intensity focus US could increase the penetration of curcumin into deep brain tissue by opening the blood−brain barrier, thereby significantly enhancing efficacy compared with only Cur-NBs group. The Cur-NBs platform is also hopeful for potential drug delivery restricted by the blood−brain barrier for central nervous system disease therapy. Besides, low-intensity focused US combined with bubble-gene complexes has been reported to achieve noninvasive targeted gene delivery for the treatment of PD. Fan et al. explored cationic MBs as gene carriers to improve the stability of the bubble-gene complex [126]. A transcranial-focused ultrasound interacted with the bubble-gene complex and promoted therapeutic gene transient permeation while inducing local expression.

In addition to PD, another common neurodegenerative disease is Alzheimer’s disease, which is characterized by neurogenesis disorder in the dorsal hippocampus with amyloid deposition and neurofibrillary tangles [127]. Because the lesion is in deep brain tissue, micro/nanobubble-assisted ultrasound is expected to overcome the disadvantage of insufficient permeability of traditional oral drugs. MBs were used to carry antibodies in combination with nanostructured polyethylene glycol-polylactic acid (PEG-PLA) to transport amyloid beta peptide into brain tissue [128]. The US irradiation of MBs caused the polymer nanomaterials to release beta peptides and target to Alzheimer’s disease biomarkers. This method can effectively improve the therapeutic effect of nanomaterials on brain diseases.

### 6.6. Other Advanced Therapeutic Applications

Micro/nanobubble-assisted ultrasound has been demonstrated in a number of new therapeutic areas, such as combination with insulin-like growth factor-1 to generate a better therapeutic response to noise-induced hearing loss [129] or association with transportation of corresponding genes or transposase to the liver of hemophilia B mice to enhance the expression level of factor IX in the liver, so as to achieve the purpose of adjuvant treatment of hemophilia [130]. US-induced delivery technology associated with superparamagnetic iron oxide (SPIO) nanoparticle-loaded gastro-retentive tablets to generate bubbles was also explored to enhance the gastric absorption of drugs and enable imaging monitoring [131]. This strategy expects to assist orally administered nanoparticles to overcome the limitations of treatment in gastric diseases such as excessive secretions, gastric wall motility and the local acidic microenvironment.

Encapsulating statins and CF680 dyes with nano-sized droplets was explored to treat degenerative disc disease [132]. Furthermore, in dental diseases, ultrasound-induced cavitation effects have been shown to effectively enhance the penetration of submicron bubbles into dentinal tubules [133]. Micro/nanobubble-assisted ultrasound-based immunotherapies such as antibody-based immunotherapy, cytokine gene therapy and dendritic cell-based vaccines have also been widely studied, especially in the fields of tumors and neurological diseases.

Another therapeutic area of micro/nanobubbles is kidney disease. This is because conventional oral drugs are generally difficult to accumulate in high concentrations in the target area. For example, the typical mechanism of preventing the development of chronic kidney disease is to inhibit the deposition of fibrin in the renal interstation. Wei et al. used polylactide-co-glycolide (PLGA) nanoparticles loaded with PPARγ agonist (rosiglitazone, RSG) to generate PLGA-RSG nanoparticles (PLNPs-RSG). PLNPs-RSG was combined with SonoVue^®^ MBs to reduce a novel complex, PLNPs-RSG-MBs (Figure 7) [134]. The synthetic complex was stimulated by ultrasound in a rat model of renal fibrosis caused by ureteral obstruction and showed an effective effect of reducing the degree of fibrosis. In addition, ultrasound combined with bubble delivery to enhance renal tubular gene expression has also been proven to be an effective method for the therapy of kidney diseases [135].

## 7. Challenges and Future Directions

Micro/nanobubble-assisted ultrasound has been evidently demonstrating high efficiency in therapeutic delivery in many preclinical studies. However, in order to realize clinical translation and take the best advantage of the unique micro/nanobubble properties, there are still several challenges that need to be overcome.

First, the biosafety in micro/nanobubbles still needs to be proved by more long-term research. Future research directions are necessary to address exact mechanisms and clarification in vivo, including the process of synergistic treatment to perfect strategies to reduce side effects. Although a variety of biomaterials have been proven to be helpful, a systematic investigation—including their pharmacological and bio-toxic dynamics, bio-distributions, degradation behaviors and detailed metabolism—is required to be elucidated for careful consideration of clinical applications. The easiest way is to adjust the size, structure and properties of bubbles on the basis of FDA-approved biomaterials. They can also be used in combination with different materials to improve their bioavailability.

Second, the composition of the bubble delivery system needs further optimization. Current bubbles suffer from insufficient specificity and low pharmacokinetics. In the future, it will be necessary to strengthen the exploration of more efficient new bubbles to optimize the drug-loading capacity and enhance drug targeting, in vivo stability, drug encapsulation and acoustic capabilities. Simplified bubble production conditions and uniform bubble size are also critical for the successfully efficient utilization of agents. Moreover, optimizing the physical properties of the bubble shell materials such as viscoelasticity or surface modification ability is also an important way to improve therapeutic efficiency. Further exploration is also highly desired toward the optimization of US irradiation.

Third, an efficient drug delivery system requires accurate imaging monitoring for intuitive visualization and quantitative evaluation to realize readjustment. Monitoring acoustic effect when bubbles are coupled with US energy can strengthen the monitoring of the micro mechanism and will be necessary to assess bubble applications. The optimization of imaging hardware, acoustic imaging parameters, and pulse sequences will also be very remarkable if achieved in future research. Multifunctional bubbles such as incorporating magnetic nanoparticles into the bubble shell may hold a high development prospect. These multifunctional bubbles can not only be applied as therapeutic agents, but also as a dual mode imaging method as, for example, photoacoustic agents. Although the combinations of US with other imaging modalities have shown great promises in translational research, more comprehensive incorporations among versatile modalities need to be further explored to improve the sensitivity of the evaluations. Furthermore, effective molecular imaging materials or integration of diagnosis and treatment are waiting for further exploration in the future.

In summary, the combination of micro/nanobubbles and US has shown a bright future in therapeutic delivery fields. Although there remain many obstacles to overcome, more functional bubble and ultrasonic system optimization will be anticipated to come into use and successfully translate micro/nanobubble-assisted ultrasound from pre-clinical to clinical in the future to provide more benefits for patients.

## Figures and Tables

**Figure 1 pharmaceutics-14-00480-f001:**
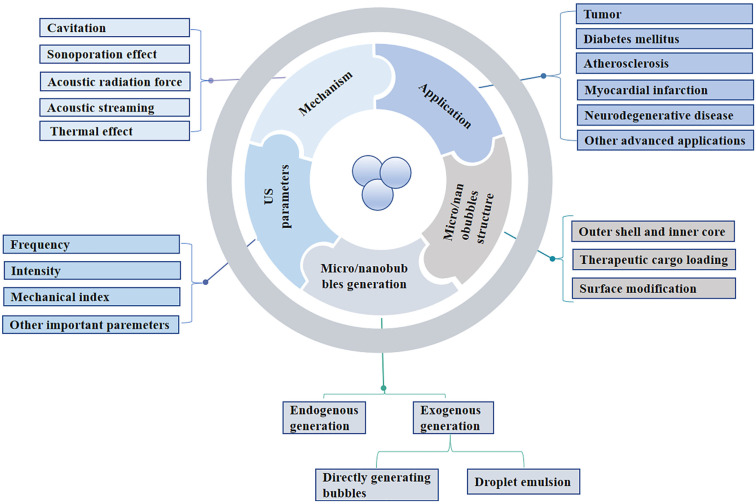
Summary of the mechanism and therapeutic delivery in micro/nanobubble-assisted ultrasound.

**Figure 2 pharmaceutics-14-00480-f002:**
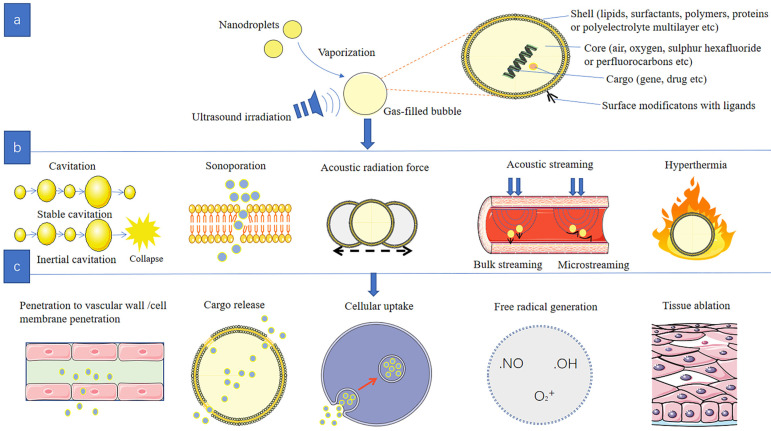
(**a**): Schematic diagram of the structure of the bubble. (**b**): Therapeutic mechanisms of bubble -assisted ultrasound. (**c**): Biological effects of bubble -assisted ultrasound.

**Figure 3 pharmaceutics-14-00480-f003:**
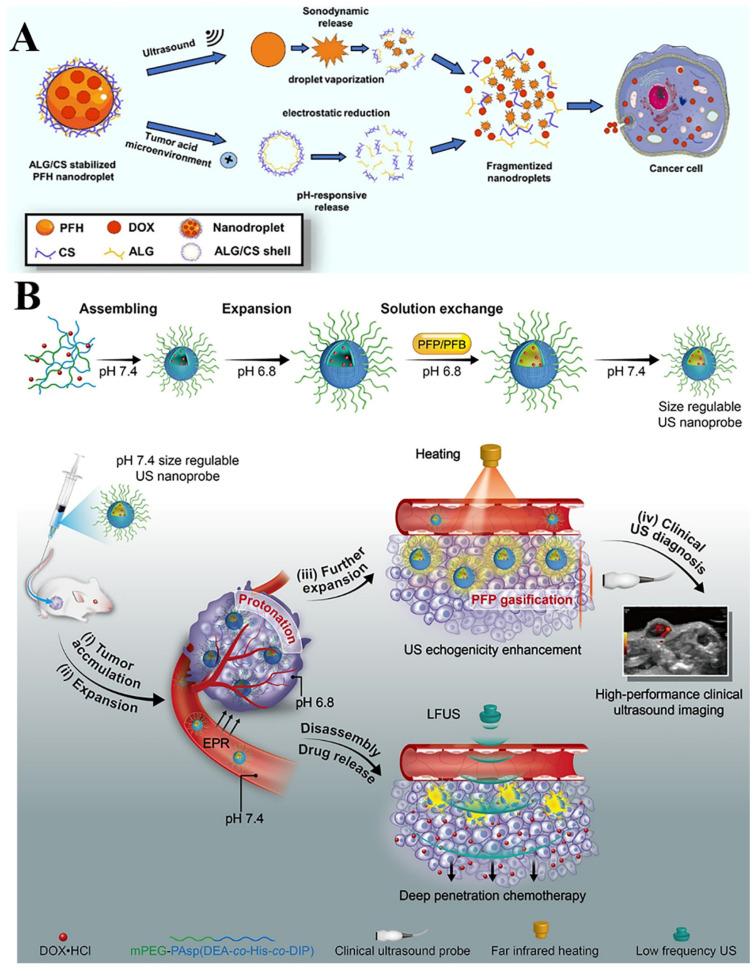
(**A**) Schematic illustrations for the construction of chitosan/alginate-stabilized perfluorohexane (PFH) nanodroplets and the ultrasound-responsive and pH-responsive drug release. Reproduced with permission from [92], Int J Biol Macromol, 2021. (**B**) Schematic illustration of theranostic nanoprobes showing tunable size and performance in vivo derived from multi-stimulation sensitivity. Reprinted from Journal of Controlled Release. Reproduced with permission from [93], ACS Nano, 2018.

**Figure 4 pharmaceutics-14-00480-f004:**
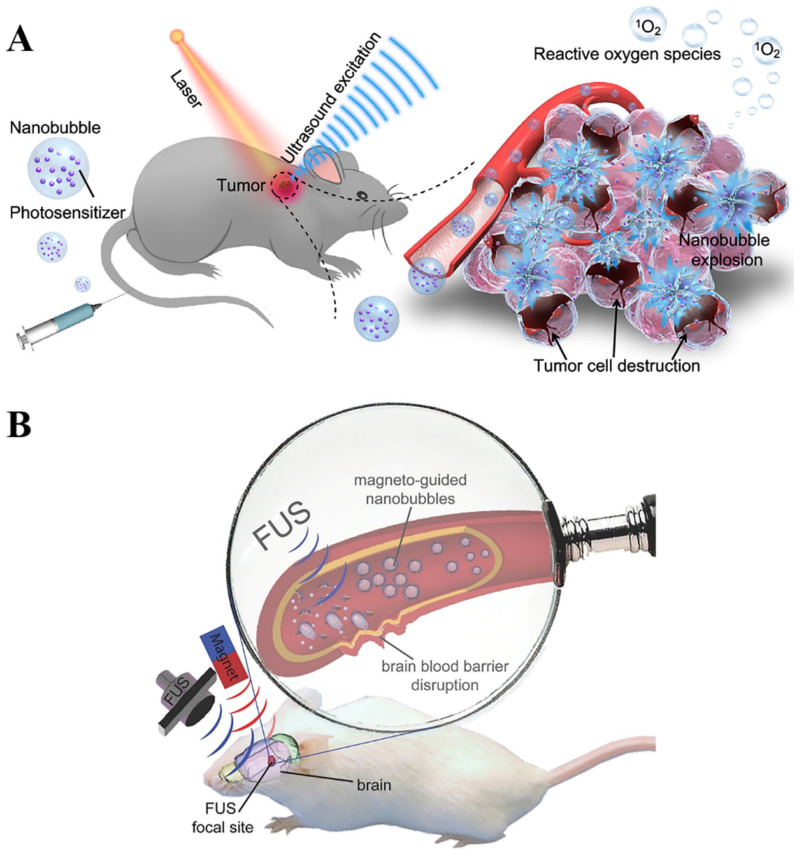
(**A**) Ultrasound-excited massive nanobubble mechanical explosion with photodynamic therapy, leading to tumor cell destruction. Reproduced with permission from [100], ACS Appl Mater Interfaces, 2021. (**B**) Schematic diagram of magnetic guidance combined with nanobubble and focused ultrasound. Reproduced with permission from [101], Adv Mater, 2015.

**Figure 5 pharmaceutics-14-00480-f005:**
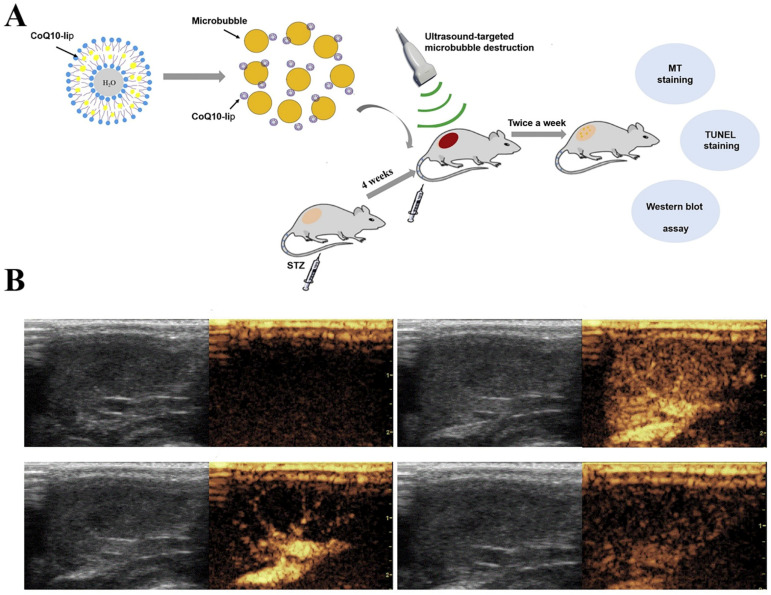
(**A**) Schematic illustrations of coenzyme Q10-loaded liposomes (CoQ10-lip) combined with ultrasound-targeted microbubbles to treat early diabetic nephropathy. (**B**) The process of ultrasound-targeted microbubble destruction. Reproduced with permission from [107], Int J Pharm, 2017.

**Figure 6 pharmaceutics-14-00480-f006:**
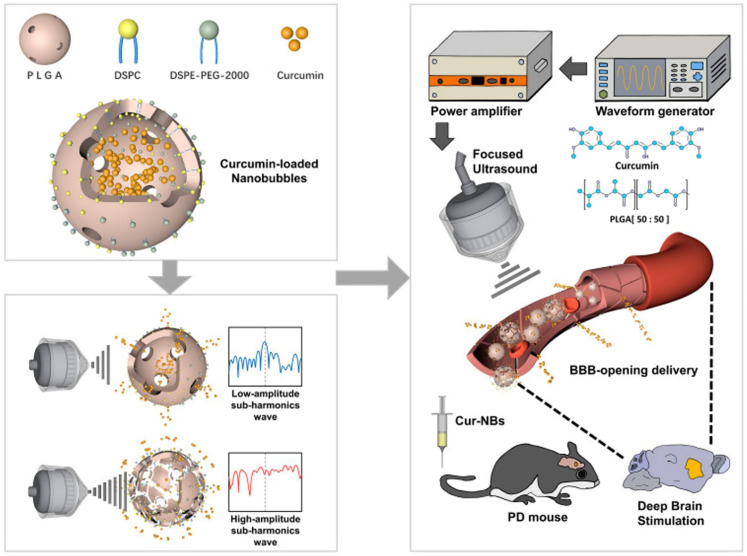
Schematic illustration of the noninvasive localized delivery of curcumin-loaded Lipid-PLGA hybrid nanobubbles (Cur-NBs) combined with low-intensity focus ultrasound for PD therapy. Reproduced with permission from [125], Int J Nanomedicine, 2021.

**Figure 7 pharmaceutics-14-00480-f007:**
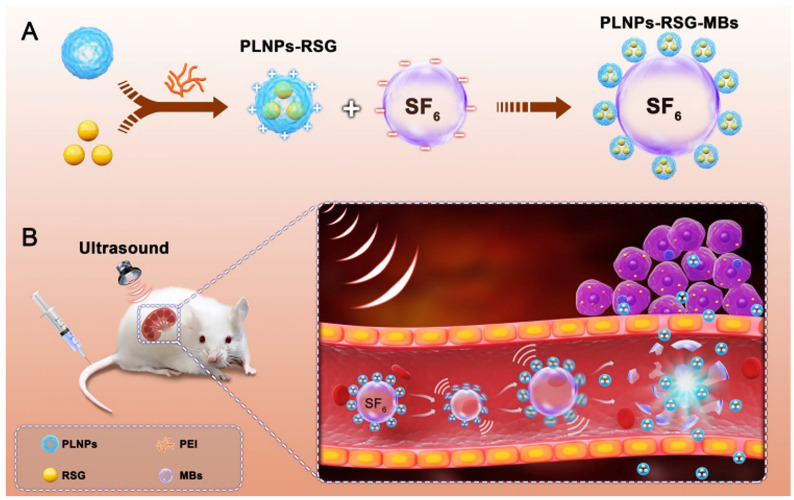
(**A**) Schematic illustration of PLNPs-RSG-MBs complex synthesis. (**B**) Schematic illustration of improved drug delivery to a UUO rat kidney due to the combination of the PLNPs-RSG-MBs complex with US exposure. Reproduced with permission from [134], Int J Nanomedicine, 2020.

## Data Availability

Not applicable.

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
