# Peer review of "Mechanistic Insights and Therapeutic Delivery through Micro/Nanobubble-Assisted Ultrasound"

_pharmaceutics, 2022, doi:10.3390/pharmaceutics14030480_

Round 1

Reviewer 1 Report

My comments and suggestions are summarized in the attached file.

Author Response

Title: " Mechanistic Insights and Therapeutic Delivery through Micro/Nano-Bubbles-Assisted Ultrasound "

Manuscript ID: pharmaceutics-1575036

We appreciate the editor and reviewers for the helpful suggestions. We have included a point-by-point response to address the comments from the reviewers, and made a thorough revision in the updated manuscript. New revisions have included additional information and increased the readability of the manuscript.

Reviewer's comments:

This review manuscript present and discuss the use of ultrasound fields with the generations of nano and micro bubbles for therapeutic drug delivery applications. Furthermore, the paper summarize some potential biomedical applications of ultrasound assisted micro and nano bubbles. The manuscript is well written and the structure and images are convincing. Therefore, I would like to better understand what is the major contribution of this manuscript compared with the one entitled Ultrasound and microbubble mediated therapeutic delivery: Underlying mechanisms and future outlook (2020)? I have others few comments that I believe would strengthen the paper and thus need to be addressed before the possible acceptance.

In summary, in my judgment the manuscript brings a good review about the use of ultrasound fields in therapeutic drug delivery. It’s a hot research area with many important aspects to be done. But, the authors should explain all aspects mentioned above for the manuscript be accepted.

Response: In the review entitled Ultrasound and microbubble mediated therapeutic delivery: Underlying mechanisms and future outlook (2020), the therapeutic strategy of using ultrasound for improved drug delivery are summarized with the special focus on cancer therapy.

Compared with the mentioned literature, this manuscript not only introduces ultrasound combined with microbubbles, but also elaborates the value of nanobubbles in therapeutic delivery. Besides, this manuscript has supplemented the drug loading methods and the ligand modification of the bubble surface. In addition to applications in tumor therapy, this manuscript has emphatically introduced potential applications of micro/nano-bubble-assisted ultrasound in diabetes mellitus, cardiovascular disease, neurodegenerative diseases and other advanced therapeutic diseases.

  1. In page 3, figure 2, the legend and/or the text should contain a better explanation about the figure. The figure have letters a, b and c, and I didn’t found any citation along the text. For example, in page 6, line 187.

Response: According to your suggestion, we added the following text about the figure 2 in the revised manuscript:

“Figure 2. Schematic diagram of the structure of bubble and its therapeutic mechanism and biological effect in combination with ultrasound.”

Citations about letters a, b and c of the figure have been supplemented in the revised manuscript (see P 3, line 105 and P 7, line 262).

  1. In page 5, section IV, the US section should be improved. The present section presents a poor description about the ultrasonic parameters that should be optimized to achieve a good efficiency.

Response: We agree with the reviewer’s assessment. Accordingly, throughout section IV, in page 6-7, We supplemented the following text in the revised manuscript:

“Any one of these ultrasound parameters will largely affect drug delivery processes by influencing bubble−cell interactions. It is valuable to modulate responses by changing ultrasound conditions through estimating the impact of ultrasound settings on the physiologic process involved in ultrasound induced drug delivery[1].”

“The cavitation behavior of bubbles at a certain frequency will mainly depend on their size, as bubble response will be higher around their resonant radius[2].Most studies report an ultrasound center frequency of around 1 MHz because it roughly matches the resonance frequency of the majority of bubbles (volume-weighted) in standard microbubble formulations (±3 μm in size)[3]

“Different from high intensity focused ultrasound (HIFU) which can thermally ablate tissues via hyperthermia in different carcinomas at 1000 W/cm2 [4], low intensity ultrasound (US) defined as therapeutic US, with a relatively lower intensity than HIFU, has a great potential in apoptosis therapy for cancer and can be relatively easily applied[5].”

“For example, while bubble plays a significant role in the process of US-induced blood–brain barrier (BBB) opening, the level of cavitation involved in the bubble-ultrasound interaction must be evaluated to guarantee BBB-opening level and quality. The mechanical index as the major indicators in gauging inertial cavitation activity, it would be valuable to understand the roles of cavitation sources on ultrasound-induced BBB-opening[6].”

“Besides, It is reported that the ultrasound pulse length can have a major impact as well. Compared with long ultrasound pulses (ms to s), it seems that very short pulses (few µs) might be more efficient in combination with high acoustic pressures[7]. Karshafian et al. reported that cell permeability increased and viability decreased with increasing peak negative pressure, pulse repetition frequency, pulse duration and insonation time and with decreasing pulse centre frequency. But cell permeability and viability did not correlate with bubble disruption. The results indicated that ultrasound exposure parameters can be optimized for therapeutic sonoporation and that bubble disruption is a necessary but insufficient indicator of ultrasound-induced permeabilisation[8].”

  1. In page 6, line 220, the authors should cite some recent works about primary and secondary forces. For example:

Primary forces:

Interparticle acoustic radiation force between a pair of spherical particles in a liquid exposed to a standing bulk acoustic wave

Secondary force:

Acoustic interaction forces and torques acting on suspended spheres in an ideal fluid; Numerical determination of the secondary acoustic radiation force on a small sphere in a plane standing wave field.

Response: According to your suggestion, we have cited the recent works about primary and secondary forces (see P 8, line 302 and line 304).

Reference

  1. Man, VH.; Truong, PM.; Li, MS.; Wang, J.; Van-Oanh, N-T.; Derreumaux, P.; Nguyen, PH. Molecular Mechanism of the Cell Membrane Pore Formation Induced by Bubble Stable Cavitation. The Journal of Physical Chemistry B 2019, 123, 71-78.
  2. Wu, J.; Nyborg, WL. Ultrasound, cavitation bubbles and their interaction with cells. Advanced Drug Delivery Reviews 2008, 60, 1103-1116.
  3. Roovers, S.; Segers, T.; Lajoinie, G.; Deprez, J.; Versluis, M.; De Smedt, SC.; Lentacker, I. The Role of Ultrasound-Driven Microbubble Dynamics in Drug Delivery: From Microbubble Fundamentals to Clinical Translation. Langmuir 2019, 35, 10173-10191.
  4. Wang, X-J.; Yuan, S-L.; Lu, Y-R.; Zhang, J.; Liu, B-T.; Zeng, W-F.; He, Y-M.; Fu, Y-R. Growth inhibition of high-intensity focused ultrasound on hepatic cancer in vivo. World J Gastroenterol 2005, 11, 4317-4320.
  5. Feng, Y.; Tian, Z-M.; Wan, M-X.; Zheng, Z-B. Low intensity ultrasound-induced apoptosis in human gastric carcinoma cells. World J Gastroenterol 2008, 14, 4873-4879.
  6. Chu, P-C.; Chai, W-Y.; Tsai, C-H.; Kang, S-T.; Yeh, C-K.; Liu, H-L. Focused Ultrasound-Induced Blood-Brain Barrier Opening: Association with Mechanical Index and Cavitation Index Analyzed by Dynamic Contrast-Enhanced Magnetic-Resonance Imaging. Scientific Reports 2016, 6, 33264.
  7. Fan, Z.; Chen, D.; Deng, CX. Improving ultrasound gene transfection efficiency by controlling ultrasound excitation of microbubbles. Journal of Controlled Release 2013, 170, 401-413.
  8. Karshafian, R.; Bevan, PD.; Williams, R.; Samac, S.; Burns, PN. Sonoporation by Ultrasound-Activated Microbubble Contrast Agents: Effect of Acoustic Exposure Parameters on Cell Membrane Permeability and Cell Viability. Ultrasound in Medicine & Biology 2009, 35, 847-860.

Reviewer 2 Report

The manuscript is an example of how Ultrasound with low frequency assisted drug delivery has been widely investigated as a non-invasive method to enhance the permeability and retention effect of drugs.

The following aspects should be addressed:

1-An index before introduction including all sections should be helpful

2-membrane permeability or permeability and retention should be included in keywords

3-in fig 2, O2 (-) is not a radical and should be removed and tissue ablaton should be corrected

4-Introduction seems to have little information, pleas einclude a general perspective of the area of study and include more references.

5-different section for inner and outer shell is not necessary, please reunify

6-the following hydrogen (H2), nitrogen (N2), oxygen (O2), are incorrectly written

7-please insert and explain more examples of section 2.4 and 3

8-there are so many figures included from other publications with different settings

9-Please include more details about safety issues in a separated section

Author Response

Title: " Mechanistic Insights and Therapeutic Delivery through Micro/Nano-Bubbles-Assisted Ultrasound "

Manuscript ID: pharmaceutics-1575036

We appreciate the editor and reviewers for the helpful suggestions. We have included a point-by-point response to address the comments from the reviewers, and made a thorough revision in the updated manuscript. New revisions have included additional information and increased the readability of the manuscript.

Reviewer's comments:

The manuscript is an example of how Ultrasound with low frequency assisted drug delivery has been widely investigated as a non-invasive method to enhance the permeability and retention effect of drugs.

The following aspects should be addressed:

  1. An index before introduction including all sections should be helpful

Response: According to your suggestion, we added the following index in the revised manuscript.

Index

Introduction 02

Micro/nanobubbles structure 03

Outer shell and inner core 04

Therapeutic cargo loading 04

Surface modification 05

Micro/nanobubbles generation 05

Endogenous generation 06

Exogenous generation 06

US parameters 06

Mechanisms of micro/nanobubbles-assisted drug delivery 07

Cavitation effect 07

Sonoporation effect 08

Acoustic radiation force 08

Acoustic streaming 08

Thermal effect 09

Biomedical applications 09

Tumor therapy 09

Diabetes mellitus 12

Atherosclerosis 14

Myocardial infarction 15

Neurodegenerative disease 15

Other advanced therapeutic applications 17

Challenges and future directions 19

Acknowledgements 20

References. 20

  1. membrane permeability or permeability and retention should be included in keywords

Response: As suggested by the reviewer, we have included membrane permeability in keywords.

  1. in fig 2, O2 (-) is not a radical and should be removed and tissue ablaton should be corrected

Response: Thanks a lot for the reviewer for pointing out the errors, we have removed O2 (-) in fig 2 and corrected tissue ablation spelling.

  1. Introduction seems to have little information, please einclude a general perspective of the area of study and include more references.

Response: Thank you for your suggestion. We added the following text in the revised manuscript.

“To overcome this limitation, nanobubbles (NBs) have been investigated for ultrasound triggered drug release outside the bloodstream, which have been known to exist in commercially available MB formulations such as Definity® (phospholipid-coated per-fluoropropane filled bubbles, mean size 1.1 µm – 3.3 µm, each mL contains a maximum of 1.2 × 1010 perflutren lipid microspheres)[1]. The other currently marketed product is Optison®, microbubbles stabilized by human serum albumin with perflutren (mean size 3.0-4.5µm, each mL contains 5.0-8.0×108 protein-type A microspheres).”

“Although NBs provide US contrast enhancement at frequencies below their resonance and hence provide promise for diagnostic use, they are still in the preclinical stage[2].”

  1. different section for inner and outer shell is not necessary, please reunify

Response: As suggested by the reviewer, we have reunified the two different sections for inner and outer shell. Accordingly, we have updated this section in the index and fig 1.

  1. the following hydrogen (H2), nitrogen (N2), oxygen (O2), are incorrectly written

Response: Following the reviewer’s suggestion, we have corrected the spelling of hydrogen (H2), nitrogen (N2), oxygen (O2) (see page 5, line 134 and line 135).

  1. please insert and explain more examples of section 2.4 and 3

Response: To address the reviewer’s concern, more explanations of section 2.4 and 3 have been included in the revised manuscript. We supplemented the following text in the revised manuscript.

“Another typical example is that glycosylation has been applied for both macromolecular and liposomal carriers for cell-selective drug targeting, the receptor-mediated targeting by glycosylation has been used for therapy with nucleic acids[3]

“In general, there are three ways of generating bubbles. The most common class uses compression of the air stream to dissolve air into liquid, which is subsequently released through a specially designed nozzle system, to nucleate small bubbles as potentially nanobubbles, based on the cavitation principle. The second class uses power ultrasound to induce cavitation locally at points of extreme rarefaction in the standing ultrasonic waves. The third class uses an air stream delivered under low offset pressure[4]

Specific references in the revised manuscript are listed as follows:

“Hirsjarvi, S.; Passirani, C.; Benoit, J-P. Passive and Active Tumour Targeting with Nanocarriers. Current Drug Discovery Technologies 2011, 8, 188-196.”

“Endo-Takahashi, Y.; Ooaku, K.; Ishida, K.; Suzuki, R.; Maruyama, K.; Negishi, Y. Preparation of Angiopep-2 Peptide-Modified Bubble Liposomes for Delivery to the Brain. Biological and Pharmaceutical Bulletin 2016, 39, 977-983.”

“Kawakami, S.; Hashida, M. Glycosylation-mediated targeting of carriers. J Control Release 2014, 190, 542-555.”

“Zimmerman, WB.; Tesar, V.; Butler, S.; Bandulasena, HCH. Microbubble Generation. Recent Patents on Engineering 2008, 2, 1-8.”

  1. there are so many figures included from other publications with different settings

Response: Thanks for your constructive suggestion, which is highly appreciated. Our manuscript does include multiple figures from other publications. Considering that the contents of mechanisms in fig 6 and fig 2 are partially duplicated, we have removed fig 6.  

  1. Please include more details about safety issues in a separated section

Response: Following the reviewer’s suggestion, safety issues have been discussed. We added the following text in the revised manuscript.

“First, the biosafety in micro/nanobubbles still needs to be proved by more long-term research. Future research directions are necessary to address exact mechanisms and clarification in vivo, including the process of synergistic treatment to perfect strategies to reduce side effects. Although a variety of biomaterials have been proved to be helpful, a systematic investigation including their pharmacological and bio-toxic dynamics, bio-distributions, degradation behaviors and detailed metabolism is required to be elucidated for a careful consideration into clinical applications. The easiest way is to adjust the size, structure and properties of bubbles on the basis of FDA-approved biomaterials. It can also be used in combination with different materials to improve its bioavailability”

Reference

  1. Unger, EC.; Porter, T.; Culp, W.; Labell, R.; Matsunaga, T.; Zutshi, R. Therapeutic applications of lipid-coated microbubbles. Advanced Drug Delivery Reviews 2004, 56, 1291-1314.
  2. JafariSojahrood, A.; Nieves, L.; Hernandez, C.; Exner, A.; Kolios, MC. Theoretical and experimental investigation of the nonlinear dynamics of nanobubbles excited at clinically relevant ultrasound frequencies and pressures: The role oflipid shell buckling. Paper presented at: 2017 IEEE International Ultrasonics Symposium (IUS); 6-9 Sept. 2017, 2017.
  3. Kawakami, S.; Hashida, M. Glycosylation-mediated targeting of carriers. Journal of Controlled Release 2014, 190, 542-555.
  4. Zimmerman, WB.; Tesar, V.; Butler, S.; Bandulasena, HCH. Microbubble Generation. Recent Patents on Engineering 2008, 2, 1-8.

Reviewer 3 Report

Pharmaceutics Review for Manuscript 1575036

This manuscript reviews the concepts of ultrasound-assisted drug delivery with bubble agents, both microbubbles and nanobubbles. Firstly, the authors will need to get this manuscript edited professionally by a native English speaker. There are many grammatical and typographical errors that will need to be corrected before publication.

In terms of the scientific content of the manuscript, I think that the second half of paper (section 6) dealing with the applications of ultrasound-microbubble treatments is well done. However, I do have some issues with the rest of the manuscript, which deal with the bubble synthesis, vibration physics/acoustic parameters and mechanisms. This section was not done very thoroughly, and in my opinion will need to be re-written.

Please see comments below, thank you:

Abstract:

  1. The authors write “ultrasound with low frequency” – please quantify what frequency range is meant by low frequency.
  2. What is drug tracing?
  3. “Furthermore…” this sentence needs to be re-written. What is an ultrasound-mediated bubble?
  4. Please re-write the abstract – there are many typographical and grammatical errors

Intro:

Line 48: reference needed

Line 51: I don’t believe there is any evidence that nanobubbles extravasate – it’s just a hypothesis?

Nanobubbles are relatively new and not well established. This manuscript incorporates them as if they are. Might be useful to clarify that nanobubbles have not really been validated as robust and repeatable ultrasound contrast agent, at least not nearly as much as microbubbles.

Figure 1 and Figure 2 convey the same information. Suggest removing one of the entirely. Figure 6 also has similar content.

Figure 1: Top left states that ‘sonoporation effect’ is a mechanism. For what? What causes sonoporation in the first place? Sonoporation is not a physical mechanism, it’s the result of a physical mechanism. This figure needs to be made more clear and the authors need to fix typos (or delete it entirely).

Microbubble-assisted ultrasound make sense, but there is no ‘s’ after microbubble. This is also the case in the title of the manuscript. Please correct this.

Section 2:

MNBs is an unusual acronym for the field. Consider using more traditional acronyms.

There is no real dedicated discussion of bubble dynamics. Throughout the text, there are occasional, very vague descriptions, but in my opinion this is insufficient and can be misleading. As one example, line 85: “the hard shell will not vibrate.”  What is ‘hard’ shell? Bubbles with stiffer shells will still vibrate at some point, but with different thresholds, etc etc. These sentences (throughout the manuscript) are misleading and need to be corrected in order to avoid confusion.

Please discuss the commercially available agents. What is their composition, size distribution, concentration, etc.

Section 4 is very incomplete. Please re-write this section.

Section 5 is also incomplete. The subsections, which talk about some mechanisms, are very short. If the authors do not want to focus on these, the section should be removed.

Author Response

Title: " Mechanistic Insights and Therapeutic Delivery through Micro/Nano-Bubbles-Assisted Ultrasound "

Manuscript ID: pharmaceutics-1575036

We appreciate the editor and reviewers for the helpful suggestions. We have included a point-by-point response to address the comments from the reviewers, and made a thorough revision in the updated manuscript. New revisions have included additional information and increased the readability of the manuscript.

Reviewer's comments:

This manuscript reviews the concepts of ultrasound-assisted drug delivery with bubble agents, both microbubbles and nanobubbles. Firstly, the authors will need to get this manuscript edited professionally by a native English speaker. There are many grammatical and typographical errors that will need to be corrected before publication.

In terms of the scientific content of the manuscript, I think that the second half of paper (section 6) dealing with the applications of ultrasound-microbubble treatments is well done. However, I do have some issues with the rest of the manuscript, which deal with the bubble synthesis, vibration physics/acoustic parameters and mechanisms. This section was not done very thoroughly, and in my opinion will need to be re-written. Please see comments below, thank you:

Response: Thank you. A native English speaker was invited to help us to revise the manuscript and the grammatical and typographical errors were corrected.  Following the reviewer’s suggestion, we supplemented the content of bubble generation, ultrasound parameters and mechanisms. See page 5 line 170- page 9 line 332.

Abstract:

The authors write “ultrasound with low frequency” – please quantify what frequency range is meant by low frequency.

Response: Low frequency refers to the frequency range of 20–100 kHz. We added “Ultrasound with low frequency (20–100 kHz)” in the revised manuscript.

What is drug tracing?

Response: Thanks for the question on drug tracing. To express more accurately, we modified as “The functional micro/nanobubbles loaded with drugs could provide an unprecedented opportunity for targeted delivery.” See page 1 line 15-17.

“Furthermore…” this sentence needs to be re-written. What is an ultrasound-mediated bubble?

Please re-write the abstract – there are many typographical and grammatical errors

Response: Following the reviewer’s suggestion. We revised this sentence as follows,

“Furthermore, ultrasound-mediated micro/nanobubbles destruction can effectively increase the permeability of vascular membranes and cell membranes, thereby not only increasing the distribution concentration of drugs in the interstitial space of target tissues, but also promoting the penetration of drugs through cell membranes into the cytoplasm.”

We changed “Ultrasound-mediated bubble” as “ultrasound-mediated micro/nanobubbles destruction”. All the typographical and grammatical errors in the abstract have been corrected.

Intro:

Line 48: reference needed

Response: We added a reference in page 3, line 80. The specific reference is:

“Stride, E.; Segers, T.; Lajoinie, G.; Cherkaoui, S.; Bettinger, T.; Versluis, M.; Borden, M. Microbubble Agents: New Directions. Ultrasound Med Biol 2020, 46, 1326-1343.”

Line 51: I don’t believe there is any evidence that nanobubbles extravasate – it’s just a hypothesis? Nanobubbles are relatively new and not well established. This manuscript incorporates them as if they are. Might be useful to clarify that nanobubbles have not really been validated as robust and repeatable ultrasound contrast agent, at least not nearly as much as microbubbles.

Response: We thank the reviewer for the detailed comments with nanobubbles. Some previous studies supported the notion that intact nanobubbles can extravasate and accumulate in the tumor matrix. Histology analysis further confirmed this event. Some relevant references are listed as follows:

“Hynynen, K.; Zheng, G.; Goertz, DE. Simultaneous Intravital Optical and Acoustic Monitoring of Ultrasound-Triggered Nanobubble Generation and Extravasation. Nano Lett 2020, 20, 4512-4519)”

“Pellow, C.; O'Reilly, MA.; Hynynen, K.; Zheng, G.; Goertz, DE. Simultaneous Intravital Optical and Acoustic Monitoring of Ultrasound-Triggered Nanobubble Generation and Extravasation. Nano Lett 2020, 20, 4512-4519”

“Perera, R.; Leon, AD.; Wang, X.; Ramamurtri, G.; Peiris, P.; Basilion, J.; Exner, AA. Nanobubble Extravasation in Prostate Tumors Imaged with Ultrasound: Role of Active versus Passive Targeting. Paper presented at: 2018 IEEE International Ultrasonics Symposium (IUS); 22-25 Oct. 2018, 2018; Pellow, C.; O'Reilly, MA”

Despite of this, we still agreed with the reviewer that more studies are required to explore its validation in different circumstance. To clarify it, we added a statement in the introduction section as follows, “Although NBs provide US contrast enhancement at frequencies below their resonance and hence provide promise for diagnostic use, they are still in the preclinical stage.”

Figure 1 and Figure 2 convey the same information. Suggest removing one of the entirely. Figure 6 also has similar content.

Response: Thanks a lot for the reviewer’s comment. We removed figure 6 since it was repeated. We think Figure 1 should be reserved because it summarized the main content of the full text, while figure 2 is a schematic diagram of the structure of a bubble and the therapeutic mechanisms of bubble assisted with ultrasound.

Figure 1: Top left states that ‘sonoporation effect’ is a mechanism. For what? What causes sonoporation in the first place? Sonoporation is not a physical mechanism, it’s the result of a physical mechanism. This figure needs to be made more clear and the authors need to fix typos (or delete it entirely).

Response: Sonoporation effect is that the instantaneous micropores in the cell membrane under US stimulation. Some important studies use “sonoporation effect” as a physical mechanism, expecially in the field of therapeutic delivery through Micro/Nano-Bubbles-Assisted ultrasound.

The following studies use the same statement, “Man, VH.; Truong, PM.; Li, MS.; Wang, J.; Van-Oanh, NT.; Derreumaux, P.; Nguyen, PH. Molecular Mechanism of the Cell Membrane Pore Formation Induced by Bubble Stable Cavitation. J Phys Chem B 2019, 123, 71-78”

“Lentacker, I.; De Cock, I.; Deckers, R.; De Smedt, SC.; Moonen, CT. Understanding ultrasound induced sonoporation: definitions and underlying mechanisms. Adv Drug Deliv Rev 2014, 72, 49-64”

Microbubble-assisted ultrasound make sense, but there is no ‘s’ after microbubble. This is also the case in the title of the manuscript. Please correct this.

Response: Following the reviewer’s suggestion, we removed the ‘s’ after the bubbles word.

Section 2:

MNBs is an unusual acronym for the field. Consider using more traditional acronyms.

Response: We quite agree with the reviewer that MNBs is an unusual acronym for the field. Therefore, we replace “MNBs” with “micro/nanobubbles”, according to the published literature as “Li, J.; Xi, A.; Qiao, H.; Liu, Z. Ultrasound-mediated diagnostic imaging and advanced treatment with multifunctional micro/nanobubbles. Cancer Letters 2020, 475, 92-98”.

There is no real dedicated discussion of bubble dynamics. Throughout the text, there are occasional, very vague descriptions, but in my opinion this is insufficient and can be misleading. As one example, line 85: “the hard shell will not vibrate.”  What is ‘hard’ shell? Bubbles with stiffer shells will still vibrate at some point, but with different thresholds, etc etc. These sentences (throughout the manuscript) are misleading and need to be corrected in order to avoid confusion.

Response: We are sorry for the inappropriate expression. ‘Hard-shell’ is composed of a gas core with coating materials (such as polymers or denatured proteins) whose visco-elastic property is relatively lower. Generally, hard shelled bubbles are relatively stable even under higher-intensity US irradiation and show a longer circulation time in vivo. Inversely, ‘soft-shell’ is composed monolayer coating with surfactant molecules such as palmitic acid or phospholipids.

Soft-shelled bubbles display a high degree of sensitivity when exposed to pressure change. This property of such bubbles can be ascribed to no coating or a monolayer coating with surfactant molecules such as palmitic acid or phospholipids. Due to their great compressibility, soft-shelled bubbles have high echogenicity. In order to make the expression more accurate, we modified the sentence as:

“Therefore, when the pressure change is small, the soft shell will break, but the hard shell will not.”

Please discuss the commercially available agents. What is their composition, size distribution, concentration, etc.

Response: Thanks for the advice. We supplemented the following content in the revised manuscript. “To overcome this limitation, nanobubbles (NBs) have been investigated for ultrasound triggered drug release outside the bloodstream, which have been known to exist in commercially available MB formulations such as Definity® (phospholipid-coated per-fluoropropane filled bubbles, mean size 1.1 µm – 3.3 µm, each mL contains a maximum of 1.2 × 1010 perflutren lipid microspheres)[1]. The other currently marketed product is Optison®, microbubbles stabilized by human serum albumin with perflutren (mean size 3.0-4.5µm, each mL contains 5.0-8.0×108 protein-type A microspheres).”

Section 4 is very incomplete. Please re-write this section.

Response: Thank you for constructive advice, Accordingly, we re-written the Section 4 in the revised manuscript:

“Any one of these ultrasound parameters will largely affect drug delivery processes by influencing bubble−cell interactions. It is valuable to modulate responses by changing ultrasound conditions through estimating the impact of ultrasound settings on the physiologic process involved in ultrasound induced drug delivery[2].”

“The cavitation behavior of bubbles at a certain frequency will mainly depend on their size, as bubble response will be higher around their resonant radius[3].Most studies report an ultrasound center frequency of around 1 MHz because it roughly matches the resonance frequency of the majority of bubbles (volume-weighted) in standard microbubble formulations (±3 μm in size)[4]

“Different from high intensity focused ultrasound (HIFU) which can thermally ablate tissues via hyperthermia in different carcinomas at 1000 W/cm2 [5], low intensity ultrasound (US) defined as therapeutic US, with a relatively lower intensity than HIFU, has a great potential in apoptosis therapy for cancer and can be relatively easily applied[6].”

“For example, while bubble plays a significant role in the process of US-induced blood–brain barrier (BBB) opening, the level of cavitation involved in the bubble-ultrasound interaction must be evaluated to guarantee BBB-opening level and quality. The mechanical index as the major indicators in gauging inertial cavitation activity, it would be valuable to understand the roles of cavitation sources on ultrasound-induced BBB-opening[7].”

“Besides, It is reported that the ultrasound pulse length can have a major impact as well. Compared with long ultrasound pulses (ms to s), it seems that very short pulses (few µs) might be more efficient in combination with high acoustic pressures[8]. Karshafian et al. reported that cell permeability increased and viability decreased with increasing peak negative pressure, pulse repetition frequency, pulse duration and insonation time and with decreasing pulse centre frequency. But cell permeability and viability did not correlate with bubble disruption. The results indicated that ultrasound exposure parameters can be optimized for therapeutic sonoporation and that bubble disruption is a necessary but insufficient indicator of ultrasound-induced permeabilisation[9].”

Section 5 is also incomplete. The subsections, which talk about some mechanisms, are very short. If the authors do not want to focus on these, the section should be removed.

Response: As suggested by the reviewer, we added the following text in the revised manuscript.

“The mechanisms contributing to sonoporation are categorized according to three ul-trasound settings: i) low intensity ultrasound leading to stable cavitation of bubbles, ii) high intensity ultrasound leading to inertial cavitation with bubble collapse, and iii) ultrasound application in the absence of bubbles[10]. Using low intensity ultrasound, the endocytotic uptake of several drugs could be stimulated, while short but intense ultrasound pulses can be applied to induce pore formation and the direct cytoplasmic uptake of drugs.”

“The consequence of Bjerknes forces is that for small bubbles, they will be collected at pressure maxima and become active there, while large ones will go to pressure minima and become inactive[3]. Bjerknes force has been used to ultrasonically concentrate erythrocytes, DNA and hybrodoma cells[3].”

“Acoustic streaming plays an important role especially in low-frequency sonophoresis[11]

Reference

  1. Unger, EC.; Porter, T.; Culp, W.; Labell, R.; Matsunaga, T.; Zutshi, R. Therapeutic applications of lipid-coated microbubbles. Advanced Drug Delivery Reviews 2004, 56, 1291-1314.
  2. Man, VH.; Truong, PM.; Li, MS.; Wang, J.; Van-Oanh, N-T.; Derreumaux, P.; Nguyen, PH. Molecular Mechanism of the Cell Membrane Pore Formation Induced by Bubble Stable Cavitation. The Journal of Physical Chemistry B 2019, 123, 71-78.
  3. Wu, J.; Nyborg, WL. Ultrasound, cavitation bubbles and their interaction with cells. Advanced Drug Delivery Reviews 2008, 60, 1103-1116.
  4. Roovers, S.; Segers, T.; Lajoinie, G.; Deprez, J.; Versluis, M.; De Smedt, SC.; Lentacker, I. The Role of Ultrasound-Driven Microbubble Dynamics in Drug Delivery: From Microbubble Fundamentals to Clinical Translation. Langmuir 2019, 35, 10173-10191.
  5. Wang, X-J.; Yuan, S-L.; Lu, Y-R.; Zhang, J.; Liu, B-T.; Zeng, W-F.; He, Y-M.; Fu, Y-R. Growth inhibition of high-intensity focused ultrasound on hepatic cancer in vivo. World J Gastroenterol 2005, 11, 4317-4320.
  6. Feng, Y.; Tian, Z-M.; Wan, M-X.; Zheng, Z-B. Low intensity ultrasound-induced apoptosis in human gastric carcinoma cells. World J Gastroenterol 2008, 14, 4873-4879.
  7. Chu, P-C.; Chai, W-Y.; Tsai, C-H.; Kang, S-T.; Yeh, C-K.; Liu, H-L. Focused Ultrasound-Induced Blood-Brain Barrier Opening: Association with Mechanical Index and Cavitation Index Analyzed by Dynamic Contrast-Enhanced Magnetic-Resonance Imaging. Scientific Reports 2016, 6, 33264.
  8. Fan, Z.; Chen, D.; Deng, CX. Improving ultrasound gene transfection efficiency by controlling ultrasound excitation of microbubbles. Journal of Controlled Release 2013, 170, 401-413.
  9. Karshafian, R.; Bevan, PD.; Williams, R.; Samac, S.; Burns, PN. Sonoporation by Ultrasound-Activated Microbubble Contrast Agents: Effect of Acoustic Exposure Parameters on Cell Membrane Permeability and Cell Viability. Ultrasound in Medicine & Biology 2009, 35, 847-860.
  10. Lentacker, I.; De Cock, I.; Deckers, R.; De Smedt, SC.; Moonen, CTW. Understanding ultrasound induced sonoporation: Definitions and underlying mechanisms. Advanced Drug Delivery Reviews 2014, 72, 49-64.
  11. Peng, H-m.; Zhu, P-c.; Lu, P-h. Acoustic streaming simulation and analyses in in vitro low frequency sonophoresis. Sensors and Actuators A: Physical 2017, 263, 744-753.

Round 2

Reviewer 1 Report

The authors answered all my questions. I have no more comments about the manuscript structure and results.

Reviewer 3 Report

I'd like to thank the authors for their replies. I just want to point out that significant English grammer errors are still present (for exampe, line 102 micro/nanobubbles stucture does not make any sense, as there should not be an 's' after bubble - something I pointed out last round of revision that was not addressed).

Other than that, I am satisfied. Thank you